# Comparative Root Transcriptomics Provide Insights into Drought Adaptation Strategies in Chickpea (*Cicer arietinum* L.)

**DOI:** 10.3390/ijms21051781

**Published:** 2020-03-05

**Authors:** Vijay Bhaskarla, Gaurav Zinta, Rebecca Ford, Mukesh Jain, Rajeev K. Varshney, Nitin Mantri

**Affiliations:** 1The Pangenomics Group, School of Science, RMIT University, Melbourne 3083, Australia; v.baaskarla@gmail.com; 2Shanghai Center for Plant Stress Biology, Center of Excellence for Molecular Plant Sciences, Chinese Academy of Sciences, Shanghai 200032, China; gzinta@gmail.com; 3School of Natural Sciences, Environmental Futures Research Institute, Griffith University, Brisbane, QLD 4111, Australia; rebecca.ford@griffith.edu.au; 4School of Computational & Integrative Sciences, Jawaharlal Nehru University, New Delhi 110067, India; mjain@jnu.ac.in; 5International Crops Research Institute for the Semi-Arid Tropics, Hyderabad 502324, India

**Keywords:** chickpea, drought, root, abiotic stress, gene expression, hormone, signaling

## Abstract

Drought adversely affects crop production across the globe. The root system immensely contributes to water management and the adaptability of plants to drought stress. In this study, drought-induced phenotypic and transcriptomic responses of two contrasting chickpea (*Cicer arietinum* L.) genotypes were compared at the vegetative, reproductive transition, and reproductive stages. At the vegetative stage, drought-tolerant genotype maintained higher root biomass, length, and surface area under drought stress as compared to sensitive genotype. However, at the reproductive stage, root length and surface area of tolerant genotype was lower but displayed higher root diameter than sensitive genotype. The shoot biomass of tolerant genotype was overall higher than the sensitive genotype under drought stress. RNA-seq analysis identified genotype- and developmental-stage specific differentially expressed genes (DEGs) in response to drought stress. At the vegetative stage, a total of 2161 and 1873 DEGs, and at reproductive stage 4109 and 3772 DEGs, were identified in the tolerant and sensitive genotypes, respectively. Gene ontology (GO) analysis revealed enrichment of biological categories related to cellular process, metabolic process, response to stimulus, response to abiotic stress, and response to hormones. Interestingly, the expression of stress-responsive transcription factors, kinases, ROS signaling and scavenging, transporters, root nodulation, and oxylipin biosynthesis genes were robustly upregulated in the tolerant genotype, possibly contributing to drought adaptation. Furthermore, activation/repression of hormone signaling and biosynthesis genes was observed. Overall, this study sheds new insights on drought tolerance mechanisms operating in roots with broader implications for chickpea improvement.

## 1. Introduction

Drought imposes a serious threat to crop growth and productivity [1]. Moreover, the intensity and frequency of drought periods are predicted to increase in the future climate [2]. Feeding the world’s growing population under water limiting conditions will be a big challenge. To sustain agricultural productivity, it is essential to identify crop adaptive traits/genotypes conferring drought resistance and dissect the underlying gene regulatory networks.

Chickpea (*Cicer arietinum* L.) is a self-pollinated cool-season legume crop. Chickpea grains serve as a rich source of proteins and essential amino acids. The majority of chickpea is cultivated in rainfed conditions of arid and semi-arid regions of the world, including south Asia and sub-Saharan Africa, where leftover soil moisture is the only source to fulfill plant water requirement. Chickpea productivity has been stagnated over the past decades, and terminal drought causes up to 40%–50% yield losses [3]. Water shortage during reproductive development (flower and pod formation) leads to a drastic decline in crop yields [4,5]. Terminal drought is a major constraint for chickpea growth and negatively affects its productivity [6]. Moreover, the predicted rise in the earth’s global temperature is likely to aggravate drought effects, fostering a decline in chickpea yield globally [7]. Thus, to meet global chickpea consumption of the future, there is an urgent need to identify or develop high-yielding chickpea varieties with improved drought tolerance [8].

Drought stress induces complex morphological, physiological, biochemical, and molecular alterations in the shoot and root systems of the plant. For instance, stomatal pores present on the leaf surface are closed during soil water deficit to minimize transpirational water loss [9]. Such processes are under the tight control of hormonal (e.g., ABA, abscisic acid) and genetic pathways [10]. Likewise, several root-specific modifications occur that involve an increase in root depth, biomass, and density, thus facilitating efficient water uptake under terminal drought conditions [11,12,13,14]. Increased root length density (RLD) during the vegetative stage allows the mining of water from deeper soils at the reproductive growth stage [15,16]. Sustained water availability is vital for reproductive success and contributes to crop grain yield [17,18]. The root system traits play a pivotal role in drought stress tolerance in plants and provide opportunities for crop selection and breeding. 

Drought stress leads to global transcriptional reprogramming, which involves several transcription factors and signaling pathways [19,20]. Implementation of genome-wide transcriptome analysis techniques provide holistic insights on gene regulatory networks and help to identify stress-responsive genes. These genes could serve as potential targets to enhance drought stress tolerance via plant breeding and/or gene modification technologies. Several methods, such as expressed sequence tags (ESTs) [21,22,23], superSAGE [24], and microarray [25,26], have been previously used to elucidate transcriptional responses against abiotic stresses in chickpea. These studies highlight that expression patterns of multiple genes related to various biological pathways are altered in response to abiotic stresses. With the advent of next-generation sequencing (NGS) technologies and the availability of chickpea reference genomes [27,28], now it is possible to analyze genome-scale gene expression patterns in chickpea [29,30]. RNA-seq technology allows non-targeted, in-depth investigations on transcriptional regulation underpinning developmental progression and stress adaption in plants. Recently, RNA-seq was used to generate a comprehensive gene expression atlas associated with chickpea growth and development [31], and to study drought triggered transcriptional responses in leaf and/or root organs [32,33]. 

The effects of drought stress not only depend on duration and intensity, but also on plant developmental stage. Drought stress during the vegetative period reduces growth rate, prolongs vegetative growth, and redirects root development [11,12,13,14]. Terminal drought negatively affects the development of floral organs, thus affecting fertilization and seed set [4,5]. Yield losses reported in crops are greater when drought coincides with the reproductive phase [6]. Importantly, however, root-specific transcriptional responses induced by drought remain to be investigated at the developmental time-scale. The present study deals with phenotypic and transcriptomic analyses of two chickpea genotypes exhibiting tolerant (ICC8261) and sensitive (ICC283) responses to drought stress. The main objective of this work is to get a comprehensive view of drought adaptation mechanisms operating in chickpea roots by comparing genotypic- and developmental stage-specific (vegetative, reproductive transition, and reproductive) responses. Phenotypic analyses revealed the adaptive plasticity of traits in response to drought stress. Also, dynamic changes in gene expression patterns related to different functional categories were observed. We discuss the possible role of these genes in drought tolerance that can facilitate the development of drought-tolerant chickpea varieties in the future.

## 2. Results

### 2.1. Phenotypic Responses

When exposed to drought stress, root-related traits such as dry weight, length, and surface area of the tolerant chickpea genotype (ICC8261) were significantly higher than the sensitive genotype (ICC283) at vegetative stage (VS) (Figure 1A–C). In contrast, at the reproductive stage (RS), root length and surface area of the sensitive genotype were higher than the tolerant genotype (Figure 1B,C). The average root diameter of the tolerant genotype was higher than the sensitive genotype at RS (Figure 1D). Root volume and relative water content (RWC) were similar in both genotypes at all growth stages (Figure 1E,F). 

Furthermore, shoot dry biomass was significantly higher in the tolerant genotype under drought stress at all developmental stages (Figure 1G). The chlorophyll content was significantly higher in the tolerant genotype during drought stress at RTS (Figure 1H). The specific leaf area (SLA) was lower in the drought-tolerant genotype at RS (Figure 1I). Overall, chickpea genotypes ICC8261 and ICC283 showed contrasting responses to drought stress, which is consistent with the previous findings suggesting that ICC8261 shows features of drought tolerance [14,16]. 

### 2.2. Transcriptional Responses and GO Analysis 

To gain mechanistic insights into regulatory mechanisms underlying the drought stress response, we did RNA sequencing. The tolerant genotype showed a total of 2161 differentially expressed genes (DEGs) in response to drought stress at the vegetative stage, out of which 1214 DEGs were upregulated and 947 DEGs were downregulated (Figure 2A). Similarly, in the sensitive genotype, 1873 DEGs were obtained, out of which 821 DEGs were upregulated and 1052 DEGs were downregulated (Figure 2A). Venn diagram analysis showed that 261 DEGs were commonly upregulated and 262 DEGs were commonly downregulated in these genotypes (Figure 2B). Interestingly, 61 DEGs showing upregulation in the tolerant genotype were downregulated in the sensitive genotype, and 20 DEGs showing downregulation in the tolerant genotype were upregulated in the sensitive genotype. 

At the reproductive stage, a total of 4109 DEGs were identified in the tolerant genotype in response to drought stress, out of which 2139 DEGs were upregulated and 1970 DEGs were downregulated (Figure 2C). In the sensitive genotype, 3772 DEGs were identified wherein 2038 were up- and 1734 were downregulated. Venn diagram analysis showed that 1163 and 1059 DEGs were commonly up- and downregulated in both the genotypes, respectively (Figure 2D). Out of these, 19 DEGs that were upregulated in the tolerant genotype were downregulated in the sensitive genotype, while 3 DEGs that showed downregulation in the tolerant genotype were upregulated in the sensitive genotype (Figure 2D). The reliability of RNA-Seq gene expression was confirmed by qRT–PCR. The fold changes obtained from RNA-seq and qRT–PCR showed good correlation with each other, thus validating the RNA-seq data (Appendix A).

Next, the functional over-representation of drought-responsive genes was performed by gene ontology (GO) analysis. GO terms classified genes into biological processes, molecular function, and cellular components (Figure 3). The most dominant terms recurring in the biological process category at vegetative and reproductive stages were ‘cellular process’, ‘metabolic process’, ‘response to stimulus’, ‘response to abiotic stress’, and ‘response to hormones’. In the molecular function category, ‘catalytic activity’, ‘binding’, ‘hydrolase’, ‘transporter activity’, and ‘antioxidant activity’ terms were present. The cellular component category contained terms such as ‘cell part’, ‘cytoplasm’, ‘intracellular’, ‘chloroplast’, and ‘cell wall’. Overall, these results suggest that roots underwent global transcriptional reprogramming during drought stress. 

### 2.3. Transcription Factor (TF) Genes 

We found that transcription factor families such as AP2/ERF, ARF, bHLH, bZIP, C2C2-Dof, C2H2-Dof, GRAS, MYB, NAC, and WRKY were differentially regulated (Appendix A). The AREB/ABF family transcription factor *ABI5* (LOC101506390), which plays role ABA-dependent signaling, was specifically upregulated (FC: 2.3↑) in the tolerant genotype at RS (Figure 4, Appendix A). Furthermore, members of CBF/*DREB* sub-family such as a drought inducible *DREB1A* (LOC101511871; FC: 1.3↑) and a cold-inducible *DREB1C* (LOC101491872; FC: 1.7↑) were upregulated in the tolerant genotype at RS (Figure 4). The AP2/ERF family TFs such as *ERF098* (LOC101502737, FC: −2.5↓) and *ERF062* (LOC101510582, FC: −8.6↓) were downregulated at VS in the tolerant and sensitive genotypes, respectively (Figure 4). However, *ERF113* (LOC101512295) was conversely regulated between tolerant (ICC8261, FC: 2.3↑) and sensitive (ICC283, FC: −4.7↓) genotypes at VS (Figure 4). The WRKY family member *WRKY33* (LOC101503578) was upregulated in the tolerant (ICC8261, FC: 5.1↑) and sensitive (ICC283, FC: 4.5↑) genotypes at VS (Figure 4). The bHLH family TFs showed upregulation in the tolerant genotype at VS and RS. For instance, at VS, *VIP1* (LOC101494682; FC: 1.1↑) and *bHLH18*-like (LOC101507246, FC: 1.61↑) were upregulated in the tolerant genotype, while *bHLH18*-like (LOC101507557) was upregulated in both tolerant (ICC8261; FC: 1.7↑) and sensitive (ICC283; FC: 2.7↑) genotypes (Figure 4). At RS, *VIP1* (LOC101494682; FC: 1.2↑), *bHLH93*-like (LOC101489960; FC: 4.1↑), *bHLH18*-like (LOC101507246, FC: 3.2↑) and *bHLH18*-like (LOC101507557; FC: 3.0↑) were strongly induced in the tolerant genotype, while sensitive genotype showed induction of only *bHLH93*-like (LOC101489960; FC: 2.3↑) and *bHLH18*-like (LOC101507557; FC: 4.6↑) (Figure 4). Moreover, the tolerant genotype showed upregulation in the expression of *C2C2-Dof* (LOC101496410, LOC101512379, LOC101496410), *MYB* (LOC101500866, LOC101509066, LOC101503477, LOC101490969, LOC101500181) and *BTB*/*POZ* and TAZ domain-containing protein 1 (LOC101498955) at VS and RS, whereas *ODORANT*1-like (LOC101500866) TF was upregulated in the sensitive genotype at RS (Figure 4).

### 2.4. Gene Expression of Protein Kinases 

Stress-responsive signal transduction pathways involve various protein kinases (PKs) such as mitogen-activated protein kinases (MAPK), calcium-dependent protein kinases (CDPK), SNF1-related kinases (SnRK), receptor-like kinases (RLK), and CBL-interacting serine/threonine-protein kinases (CIPKs). We found that mitogen-activated protein kinase kinase 1-like *MKK1* (LOC101493066) was upregulated in the tolerant genotype (ICC8261) at VS (FC: 1.9↑) and RS (FC: 2.2↑) (Figure 5, Appendix A). Similarly, members of the CDPK family such as *CDPK*-*SK5* (LOC101512548, FC: 1.4↑ at VS) and *CDPK3* (LOC101512470, FC: 1.1↑ at RS) showed increased expression in the tolerant genotype (Figure 5). Also, the expression of *CDPK4* (LOC101492192) was increased in both tolerant (ICC8261, FC: 2.5↑) and sensitive (ICC283, FC: 2.6↑) genotypes at RS (Figure 5). The expression pattern of RLKs such as *IMK2* (LOC101502785), *CRK25* (LOC101508862), and *PERK4* (LOC101511071) was upregulated in the tolerant genotype at VS, while *PERK14* (LOC101510818) and *HSL2* (LOC101509045) were upregulated in both genotypes at RS (Figure 5). In addition, differential regulation of wall-associated receptor kinases (WAKs) was observed where *WAK*-like 20 (LOC101515529) was upregulated at VS (FC: 1.5↑) and RS (FC: 1.9↑) in the tolerant genotype (Figure 5). The expression of *CIPK1*-like (LOC101510187) was specifically down-regulated at VS (FC: −9.0↓) in the sensitive genotype, while it was upregulated at RS (FC: 1.3↑) in the tolerant genotype (Figure 5).

### 2.5. ROS Detoxification System 

ROS is detoxified by various antioxidant enzymes like superoxide dismutase (SOD), ascorbate peroxidase (APX), catalase (CAT), and glutathione s-transferase (GST). In this study, *GST* genes (LOC101514835, LOC101508320, LOC101503639, and LOC101494097) showed upregulation in the tolerant genotype at VS and RS (Figure 5, Appendix A). Similarly, peroxidase genes (LOC101510290, LOC101498384, and LOC101512377) showed upregulation in the tolerant genotype. However, significant downregulation of *PEROXIDASE 47* (LOC101498384) and *PEROXIDASE 72*-like (LOC101512377) was observed in the sensitive genotype at VS and RS (Figure 5). Furthermore, expression of *RBOHH* (LOC101502622) and *RBOHD* (LOC101491892) were downregulated in the tolerant genotype at VS, whereas *RBOHE* (LOC101508393) and *RBOHA* (LOC101511451) were upregulated in this genotype at RS (Figure 5).

### 2.6. Transporter Family Genes 

Transporters mediate active or passive transport of stress-induced signaling molecules such as hormones, ions, and osmolytes. ATP-dependent transporters like *ABCB15*-like (LOC101501233), *ABCB19* (LOC101512893), and *ABCC12*-like (LOC101490063) were upregulated in the drought-tolerant genotype at VS, while *ABCB15*-like (LOC101489264), *ABCB29* (LOC101493070), and *ABCG22*-like (LOC101494791) were upregulated at RS (Figure 6, Appendix A). However, in the sensitive genotype, the expression of these transporters remains unchanged or even got decreased. Furthermore, expression of sugar and nitrate transporter genes were also investigated. The bidirectional sugar transporters *N3* (LOC101510607), *SWEET1*-like (LOC101515250), and *SWEET4* (LOC101488443) were upregulated in both the genotypes during drought stress (Figure 6). The dual affinity nitrate transporters *NRT1*/*PTR FAMILY 3.1* (LOC101498251) and *NRT1*/*PTR FAMILY 5.6*-like (LOC101497328) were strongly upregulated in both the genotypes at RS (Figure 6). Water molecules are transported passively by aquaporin channel proteins in the cell. We found that aquaporin genes such as *PIP2-1*-like (LOC101488859) and *TIP2-2* (LOC101505621) were upregulated in the tolerant genotype at VS and RS, respectively (Figure 6). 

### 2.7. Root-Nodule Development Genes

Chickpea roots contain nodules that help in the establishment of a symbiotic association with nitrogen-fixing rhizobia. The expression of putative nodulation receptor kinase (LOC101507037) and calcium/calmodulin-dependent serine/threonine-protein kinase *DMI3* (LOC101513751) were downregulated in the sensitive genotype both at VS and RS; however, the tolerant genotype showed marginal downregulation at RS only (Figure 6, Appendix A). In contrast, receptor-like kinases 3 (LOC101496137) and *CLAVATA1* (LOC101488348) were upregulated at VS and RS in the tolerant genotype (Figure 6).

### 2.8. Hormone-Related Genes 

Phytohormones control plant growth and development and also mediate stress responses. The auxin biosynthesis gene, indole-3-pyruvate monooxygenase *YUCCA2* (LOC101489587), was down-regulated at VS (FC: −4.3↓) in the sensitive genotype (Figure 7, Appendix A). In contrast, auxin response factor (*ARF19*-like; LOC101498659), a gene involved in the regulation of auxin-responsive signaling was upregulated in both tolerant (FC: 1.1↑) and sensitive (FC: 1.4↑) genotypes at RS. The auxin-responsive proteins (AUX/IAA) showed genotype and developmental stage-specific responses, where *IAA26*-like (LOC101496793) was upregulated in the tolerant genotype at VS (FC: 1.2↑) and *IAA29* (LOC101498854) was upregulated in both the genotypes except the sensitive genotype at RS (Figure 7). Interestingly, three *PIN* genes, including *PIN1b* (LOC101499345), *PIN1c*-like (LOC101491826), and *PIN4* (LOC101502756), were down-regulated in the tolerant genotype at RS (Figure 7). 

Cytokinin biosynthesis gene *IPT5* (adenylate isopentenyltransferase 5; LOC101499412) was specifically downregulated in tolerant genotype at VS, while upregulated in both the genotypes at RS (Figure 7). Furthermore, *AHP4* (*Arabidopsis* histidine phosphotransfer 4; LOC101495485) that regulates cytokinin signaling was upregulated in both sensitive (ICC283; FC: 4.10↑) and tolerant genotypes (ICC8261; FC: 3.5↑) at VS and RS, respectively (Figure 7). Also, histidine kinase 5 *AHK5* (LOC101492661) was specifically downregulated in the tolerant genotype at VS (Figure 7). 

Ethylene-responsive transcription factors (ERF) such as *ERF1*-like (LOC101511846) and *ERF113*-like (LOC101512295) were upregulated in the tolerant genotype at VS (Figure 7). Also, *ERF13*-like (LOC101491020) and *ERF2*-like (LOC101498159) genes were upregulated in the tolerant genotype, whereas downregulated in the sensitive genotypes at RS (Figure 7). 

ABA metabolism gene *NCED1* (9-cis-epoxycarotenoid dioxygenase 1; LOC101492033) was strongly upregulated at VS but was downregulated at RS in the sensitive genotype. The expression of *NCED3* (9-cis-epoxycarotenoid dioxygenase 3; LOC101505040) was upregulated in the tolerant genotype at RS (Figure 7). Also, abscisic acid 8′-hydroxylase 3-like *CYP707A3*-like (LOC101503447) showed upregulation in the tolerant genotype at VS, while abscisic acid 8′-hydroxylase 1-like *CYP707A1*-like (LOC101505927) was upregulated at RS (Figure 7). Furthermore, the ABA receptor complex (*PYR*/*PYL*/*RCAR*) genes and protein phosphatase 2C (*PP2Cs*) were differentially regulated in both the genotypes. The ABA receptor genes such as *PYL4*-like (LOC101508615 and LOC101509736) were significantly downregulated in tolerant genotype at VS, while protein phosphatases 2c genes (*PP2C*; LOC101488329 and LOC101506371) were specifically upregulated in the sensitive genotype (Figure 7). However, *PYL4*-like (LOC101508615) was conversely regulated between tolerant (ICC8261, FC: 1.1↑) and sensitive (ICC283, FC: −2.9↓) genotypes at RS. 

Gibberellin 3-beta-dioxygenase genes (*GA3ox*; LOC101498534, LOC101498875) were downregulated in the tolerant genotype at VS (Figure 7). However, gibberellin 20 oxidase 2-like (*GA20ox*; LOC101491937) was upregulated in both the genotypes at RTS and RS (Figure 7). The expression of gibberellin receptor *GID1B*-like (LOC101492626) and DELLA (LOC101507839, LOC101508270, LOC101494454, LOC101513638) were downregulated at all stages in both the genotypes (Figure 7). 

Jasmonic acid pathway genes such as linoleate 9S-lipoxygenase-like (*LOX*, LOC101490986; FC: 4.5↑) and phospholipase A2-alpha-like (LOC101508624; FC: 1.7↑) genes were upregulated in the tolerant genotype at VS, but were downregulated in the sensitive genotype (FC: −2.7↓; FC: −2.5↓) (Figure 7). Furthermore, the tolerant genotype showed upregulation in the expression of linoleate 9S-lipoxygenase homologs (LOC101491298 and LOC101491624), specifically at VS and RS (Figure 7). Also, we observed increased expression of TIFY genes in the tolerant genotype at VS (LOC101495051, LOC101503507, LOC101503456) and RS (LOC101492009, LOC101495051, LOC101488350, LOC101503456, LOC101491099) (Figure 7). 

## 3. Discussion 

Drought imposes a serious threat to agriculture. The root system plays a crucial role in the uptake of water and nutrients from the soil, thus helps in coping with drought stress. Root development and architecture are determined by intrinsic genetic properties and/or modulated by various environmental factors such as water availability [32,33]. So, we compared phenotypic and root-specific transcriptional responses of two contrasting chickpea genotypes at vegetative (VS), reproductive transition (RTS), and reproductive (RS) stages. 

The tolerant genotype had higher root and shoot dry biomass under drought stress, possibly contributed by better water foraging capacity of roots and/or onset of resilience mechanisms. Indeed root length and surface area of tolerant genotype were higher at the vegetative stage; however, conservative growth was observed at the reproductive stage. In contrast, the root diameter showed the opposite trend. Longer roots help plants to forage water from deeper soils, and the larger surface area provides better opportunities for mycorrhizal colonization that facilitate nutrient acquisition [34]. The opposite response of root diameter could be explained by the fact that diameter controls the length and surface area for given biomass allocated to the root system, thus summarize the overall effect [35]. Chlorophyll content of the drought tolerant genotype was higher under drought stress at RTS, suggesting improved light-harvesting and energy production capacity to support growth and reproduction [36]. Also, lower specific leaf area in the tolerant genotype can improve water-use efficiency, as reported previously [37,38]. 

RNA sequencing allows genome-scale quantification of transcriptome changes underlying developmental transitions and stress responses [39]. In the present work, genes related to various biological functions were altered in response to drought stress. Transcription factor (TF) genes related to ABA-dependent (*ABI5*) and ABA-independent (*DREB1A* and *DREB1C*) pathways were upregulated in the tolerant genotype at RS, indicating robust induction of multiple regulatory networks to impart drought tolerance. Also, genotype-specific alterations in other TF genes such as *AP2*/*ERF*, *bHLH*, *bZIP*, *DOF*, and *WRKY* were observed. The AP2/ERF forms a large group of plant-specific TFs induced by multiple stresses and phytohormones [40,41]. *ERFs* were previously shown to be induced by wounding and biotic stresses [42]. Likewise, WRKY family TFs have been suggested to play a role in adaptation to biotic and abiotic stresses, and modulate stress-responsive signaling and ROS production [43]. WRKY33 is involved in the activation of peroxidases and glutathione S-transferases under drought stress, playing a role in scavenging ROS [44]. We found that bHLH family TF genes (*VIP1*, *bHLH18*-like, and *bHLH93*-like) were upregulated in the tolerant genotype. These TFs may contribute to improved defense; for instance, *VIP1* activates stress-responsive genes via the mitogen-activated protein kinase pathway [45]. Also, other TF genes such as *C2C2-Dof*, *MYB*, *BTB*/*POZ*, and *TAZ* were upregulated in the tolerant genotype. Overall, upregulation of TFs in the tolerant chickpea genotype may activate stress-responsive genes conferring drought tolerance.

Protein kinases are integral components of signal transduction pathways playing an important role in the perception and activation of stress-responsive pathways [46]. The *MKK1* gene was upregulated in tolerant genotype both at VS and RS upon drought exposure. It may contribute to the maintenance of cellular homeostasis by lowering ROS levels under stress [47,48]. CDPK family members (*CDPK*-*SK5*, *CDPK3*, and *CDPK4*) were upregulated in both the genotypes. CDPKs are activated by binding of intracellular Ca^2+^ with a calmodulin-like domain that eventually regulates the downstream targets [49]. For instance, *CDPK4* phosphorylates NADPH oxidase to regulate ROS production during stress [50,51,52]. We found upregulation of RLKs (*IMK2*, *CRK25*, and *PERK4*) in the tolerant genotype at VS, while *PERK14* and *HSL2* showed up-regulation in both genotypes at RS. RLKs get rapidly activated during the early stages of drought stress, as reported previously [53,54]. *PERK4* modulates root tip growth via ABA signalling, and *HSL2* regulates lateral root emergence by controlling cell separation [53,54]. The *WAK*-like 20 was upregulated in the tolerant genotype at VS, which possibly facilitate communication between the cell wall and cytoplasm, and regulate stress responses [55,56,57,58,59]. Therefore, their enhanced expression in the tolerant genotype may contribute to cell wall loosening that minimizes root growth inhibition under drought. Also, *CIPK1*-like was upregulated in the tolerant genotype at RS. The CIPK1 functions as a regulator of ABA-dependent and independent signaling pathways by interacting with calcium sensors [60]. Taken together, early and robust upregulation of protein kinases possibly contributes to drought tolerance by activation of downstream stress signaling pathways.

ROS act as both signaling and damaging molecules whose levels are determined by antioxidant molecules and enzymes [47]. The expression of many antioxidant enzymes, such as glutathione transferases (GST) and peroxidases (PER), was upregulated in the tolerant genotype. Simultaneous upregulation of *GST* and *PER* indicate that multiple ROS detoxification components efficiently scavenge ROS under drought; however, their downregulation in the sensitive genotype may lead to excessive ROS accumulation, causing root growth inhibition. The expression of *RBOHH* and *RBOHD* were downregulated in the tolerant genotype at VS, whereas *RBOHE* and *RBOHA* were upregulated at RS. Previously, downregulation of *RBOH* family genes has been reported under drought stress [61,62]. The down-regulation of *RBOHD* and *RBOHH* may cause inhibition of ABA-dependent ROS signaling, which allows marginal root growth inhibition in the tolerant genotype. Thus, the synergistic action of a battery of antioxidant enzymes confers stress tolerance in plants [63]. 

Stress-induced signaling molecules are transported via active or passive transporters. Active transport is mediated by ATP dependent transporters such as ATP-binding cassette (ABC), whereas passive transport occurs through ion-channels and carriers such as aquaporins [64,65,66,67]. ABC transporters were upregulated in the tolerant genotype at VS (*ABCB15*-like, *ABCB19*, and *ABCC12*-like) and RS (*ABCB15*-like, *ABCB29*, and *ABCG22*-like). Upregulation of ABC transporters can be linked to multiple functions, which include auxin transport, suberin formation, ABA signaling and transport, pollen development, leaf water retention, and stress tolerance [64,65,66,67]. Furthermore, sugar (*N3*, *SWEET1*-like, *SWEET4*) and nitrate (*NRT1/PTR FAMILY*) transporters were upregulated in both genotypes. The *SWEET11* facilitates sugar supply to roots in a cell-type-specific manner and participates in the establishment of symbiotic association [68]. Furthermore, *SWEET12* and *SUC2* functions are associated with the reallocation of carbohydrates to roots for efficient root development under drought [69]. Also, NRT transporters are involved in drought stress responses by controlling ABA transport [70]. Aquaporins passively transport water molecules, and we found upregulation of *PIP2-1*-like and *TIP2-2* in the tolerant genotype. Consistent with this, overexpression of *OsPIP1-3* and *SlTIP2-2* led to the maintenance of leaf water potential under drought stress in rice and tomato, respectively [71,72]. Also, ABA regulates aquaporin activity and hydraulic conductivity to maintain favorable plant water status [73]. Therefore, aquaporins could serve as ideal targets for engineering drought tolerance in chickpea.

Leguminous plants harbor nitrogen-fixing rhizobia in root-nodules, which facilitates the uptake of nutrients and water. It has been observed that Medicago with nodulated roots shows delayed leaf senescence as compared to non-nodulated plants under drought stress [74]. We also found downregulation of nodulation receptor kinase and calcium-dependent kinase *DMI3* in the sensitive genotype. *DMI3* acts upstream of nodulation-signaling pathway proteins (NSP1/NSP2) and is a regulator of Nod-factor genes [75,76]. Thus, their down-regulation in the sensitive genotype suggests a decrease in root nodulation, resulting in low nitrogen assimilation. In contrast, *LYK3* and *CLAVATA1* were upregulated in the tolerant genotype. *LYK3* regulates rhizobial infection via Nod-factor genes [77], and *CLAVATA1* regulates root length and nodulation [78]. Therefore, their upregulation may contribute to root nodulation and nitrogen assimilation under drought stress.

Cross-regulatory hormonal network modulates plant growth and stress response in plants [79]. Auxin regulates root and shoot growth under optimal and stressful conditions [80]. In this study, auxin biosynthesis gene (*YUCCA2*) was downregulated in the sensitive genotype at VS, indicating attenuation of auxin biosynthesis. Consistent with this inhibition of root biomass, length, and surface area were observed in the sensitive genotype. In contrast, *PIN* genes (*PIN1b*, *PIN1c*-like, and *PIN4*) were downregulated in the tolerant genotype at RS. PIN proteins act as auxin efflux carriers for basipetal auxin transport, which in turn controls lateral root formation [81]. Their downregulation may be associated with the reduction of root length and root surface area. Likewise, cytokinin controls lateral root, apical meristem, and vascular system development [82,83]. Cytokinin biosynthesis (*IPT5*) and signaling (*AHK5*) genes were downregulated in the tolerant genotype. It has been shown that *AHK5* inhibits root elongation via *ETR1*-dependent ABA and ethylene signaling pathways [84]. Also, earlier transcriptome studies highlight the dual role of ethylene to regulate growth and defense responses during stress [85]. We found upregulation of ethylene-responsive transcription factors (ERF) such as *ERF1*-like, *ERF113*-like, *ERF13*-like, and *ERF2*-like in the tolerant genotype. *ERFs* integrate ethylene and jasmonic acid signaling pathways [86] and participate in stress responses [87,88]. Ethylene is known to control lateral root formation via modulating auxin biosynthesis and transport [89], and root and shoot growth via cross-talk with ABA which is a central mediator of drought stress-induced signaling [90]. The expression of ABA hydroxylases (*CYP707A3*-like and *CYP707A1*-like) was upregulated in the tolerant genotype, which could catalyze oxidative degradation of ABA, maintaining optimal ABA levels during drought [91]. The *PYL* genes were downregulated in tolerant genotype, while phosphatases (*PP2C*) were upregulated in the sensitive genotype. ABA binds with PYL/PYR/RCAR receptors to inhibit the activity of PP2Cs that negatively regulate ABA signaling through repression of SnRK2. PP2C inhibition leads to SnRK2 de-repression, eventually phosphorylating and activating downstream transcription factors [92]. The *PYL4* downregulation and *CYP707A3* activation in drought-tolerant chickpea genotypes suggest inhibition of ABA signaling at VS, whereas the up-regulation of *NCED1* and *PYL4* indicate activation of ABA signaling at RS. Also, ABA accumulation is known to inhibit root growth by promoting ethylene biosynthesis [93]. Gibberellic acid induces cell proliferation and elongation growth in plants [94,95]. DELLA proteins are negative regulators of GA signaling that act downstream of GA receptors. The binding of DELLA to GA receptor GID1 leads to their degradation and activation of GA function [96]. We observed downregulation GA receptor (*GID1B*-like) and DELLA genes in both the genotypes. This indicates that repression of *GID1* and *DELLA* might result in reduced GA responses, thus attenuating root growth under drought stress. The role of lipid-derived phytohormone Jasmonate in abiotic stress tolerance is emerging [97]. We found that expression of JA biosynthesis genes such as *LOX*, *LOX* homologs, and *TIFY/JAZ* were significantly upregulated in the tolerant genotype, suggesting enhanced production of JA in this genotype. Previously, it was reported that activation of allene oxide synthase (AOS), an enzyme involved in JA biosynthesis, enhances drought tolerance in chickpea [97]. In addition, JA and ABA pathways are involved in the regulation of stomatal opening and closing to improve drought tolerance [98]. Overall, complex hormonal-crosstalk controls root growth under drought stress in chickpea. 

## 4. Materials and Methods

### 4.1. Plant Material and Experimental Set-Up

The seeds of two chickpea (*Cicer arietinum* L.) cultivars viz. ICC283 (Desi type) and ICC8261 (Kabuli type), exhibiting contrasting responses to drought stress, were procured from ICRISAT (International Crops Research Institute for the Semi-Arid Tropics) Hyderabad, India. The genotype ICC283 is drought-sensitive, while ICC8261 is a drought-tolerant genotype. This phenotypic divergence in response to drought is attributed to their root system architecture, where the susceptible ICC283 genotype possesses shallow root systems, while the tolerant ICC8261 genotype possesses prolific root systems [14,16]. The seeds of these genotypes were sown in 27-cm polypropylene pots containing 9.5 kg of soil. Pots were placed in controlled growth conditions of the day (28 °C) and night (23 °C) temperatures, and randomly assigned as well-watered (WW) and drought-stressed (DS). The optimal water level of 95% ASWF (available soil water fraction) was maintained in WW pots by irrigating pots on alternate days. DS pots were maintained at 95% ASWF at sowing and were brought to 70% ASWF by dry-down fifteen days before the sampling. The pots were weighed every day and water was added to compensate the water lost through transpiration. The experiment was performed in a 3 × 2 × 2 (three time points, two genotypes, and two treatment conditions) completely randomized block design. Root tissue was harvested for physiological and RNA-seq analyses from WW and DS treatments at 30, 50, and 70 days after sowing (DAS). These time points reflect the vegetative stage (VS; 30 DAS), the reproductive transition stage (RTS; 50 DAS), and the reproductive stage (RS; 70 DAS). Root tissue harvested for RNA-seq was immediately put in liquid nitrogen. Three biological replicates were used to carry out physiological and transcriptional analyses.

### 4.2. Phenotypic Analysis

Phenotypic root and shoot traits were quantified in the sensitive and tolerant chickpea genotypes at three developmental stages viz. VS (30 DAS), RTS (50 DAS), and RS (70 DAS). Dry root and shoot biomass were determined after drying plant material at 65 °C for 3 days in the oven. To determine root-related traits such as length, surface area, diameter and volume harvested roots were washed with water and dried between the folds of filter paper. Then images were captured to compute these parameters. Relative water content (RWC) was calculated as RWC = [(fresh weight − dry weight)/(turgid weight − dry weight)] × 100. Leaf-related traits such as chlorophyll content (SPAD chlorophyll meter) and specific leaf area (SLA) was calculated as SLA = leaf area (cm^2^)/leaf dry weight (g). All the phenotypic analyses were performed in at least three replications, and data were reported as mean and standard error (SE). Means were statistically compared by one-way ANOVA at the probability level of 5% (*p* < 0.05) by using SPSS software (IBM, New York, NY, USA).

### 4.3. RNA Extraction and Library Construction

Total RNA was extracted with an RNeasy Mini Kit (Qiagen, Hilden, Germany). RNA quantification was done by using a Nanodrop Lite spectrophotometer (Thermo Fisher Scientific, Wilmington, MA, USA) and the integrity of RNA was checked with Bioanalyzer 2100 (Agilent Technologies, Santa Clara, CA, USA). For library preparation and sequencing, RNA samples with a 260/280 absorbance ratio between 2.1 to 2.2, 260/230 absorbance ratio between 2.0 to 2.5, and RIN (RNA integrity number) of more than 7.0 were used.

The enrichment of mRNA samples was done using a MicroPoly(A)Purist Kit (Thermo Fisher Scientific, Waltham, MA, USA) by following the guidelines of the manufacturer. The RNA-seq libraries were prepared using Ion Total RNA-Seq Kit v2 (Thermo Fisher Scientific, Waltham, MA, USA) and sequenced by using an Ion Proton System (Thermo Fisher Scientific, Waltham, MA, USA). The low-quality reads and primer/adapter sequences (QUADTrim; https://bitbucket.org/arobinson/quadtrim) were removed to obtain single-end reads of length varying from 50–265 bp with an average of ~80 bp. We sequenced thirty-three libraries in total, each genotype represented by two treatments, three time-points, and three biological replicates. The three samples of tolerant genotype (ICC8261) at the reproductive transition stage (RTS) were not sequenced due to RNA degradation.

### 4.4. Transcriptome Analysis 

Sequenced reads were mapped to the Chickpea Reference Genome (*Cicer arietinum* v1.0) [28] by using Bowtie v2.2.3 [99] and Tophat v2.1.1 [100] with default settings. Then mapped reads were counted with HTSeq-count v0.61 [101], and differential expression analysis was performed by the EdgeR package [102]. This package runs in the R statistical environment, uses a count-based approach, and employs the over-dispersed Poisson model to account for both biological and technical variability. Finally, differentially expressed genes (DEGs) were selected by applying a cut-off of *p*-value ≤ 0.001, FDR ≤ 0.05, log2 fold change ≥ +1.0 and ≤ −1.0.

### 4.5. Gene Enrichment Analysis

To find out the homologous protein sequences of chickpea, genes were retrieved from the chickpea protein library (ftp://ftp.ncbi.nih.gov/genomes/Cicer_arietinum/protein/protein.fa.gz) and aligned against Arabidopsis protein sequences in the TAIR (ftp://ftp.arabidopsis.org/home/tair/Sequences/blast_datasets/TAIR10_blastsets/TAIR10_pep_20101214_updated) database. Then, to perform gene enrichment analysis, the matched TAIR locus IDs were used as a query for agriGO [103]. GO analysis categorized genes and gene products into three major categories: (1) biological processes (BP), (2) molecular function (MF), and (3) cellular Component (CC). 

## 5. Conclusions

Drought triggered genotype- and developmental stage-specific transcriptional reprogramming in chickpea roots. The drought-tolerant genotype specifically upregulated genes related to transcriptional regulation (*ERF113*, *ERF1*, *ERF13*, *VIP1*, *bHLH18*, *ABI5*, *CDF2*, *CDF3*-like, *DREB1A*, *DREB1B*-like00, *MYB46*, *MYB114*-like, *WRKY33*, and *RVE7*), kinase activity (MAPK *MKK1*, *CPK3*, *PERK4, HSL2*, and *CIPK1*-like), detoxification (*GST*, *GST-L1*-like, *DHAR3*, *PER47*, *PER72*-like), ROS signaling (*RBOHD*, *RBOHH*), transporter activity (*ABCB19*, *ABCB12*-like, *ABCB15*-like, *ABCG22*-like, *PIP2-1*, *TIP2-2*), nodulation (*LYK3*, *CLAVATA1*), and oxylipin biosynthesis (*LOX*), thus making them ideal candidate genes for enhancing drought tolerance. Also, differential gene expression patterns of phytohormone biosynthesis and signaling genes were observed, highlighting the possible role of hormonal networks in shaping root phenotype best suited for drought stress conditions. Overall, this study identifies key genes involved in drought stress adaptation with broader implications to generate climate-resilient chickpea varieties.

## Figures and Tables

**Figure 1 ijms-21-01781-f001:**
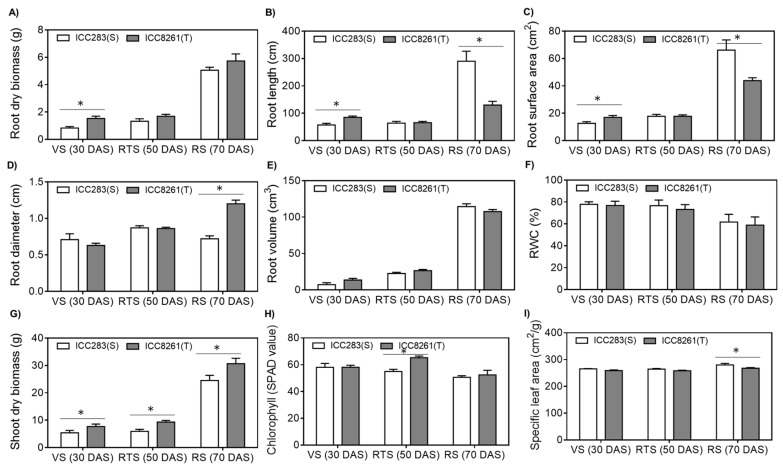
Growth and physiological responses of tolerant (T, ICC8261) and sensitive (S, ICC283) chickpea genotypes to drought stress during vegetative (VS), reproductive transition (RTS) and reproductive (RS) stages. (**A**) root dry weight (**B**) root length (**C**) root surface area (**D**) average diameter (**E**) root volume (**F**) relative water content (**G**) shoot dry weight (**H**) chlorophyll content (**I**) specific leaf area (SLA). Error bars represent standard errors (SE). Physiological data were obtained from three independent biological replicates. Statistically significant differences between sensitive and tolerant genotypes obtained by one-way ANOVA at *p* < 0.05 are depicted by an asterisk (*).

**Figure 2 ijms-21-01781-f002:**
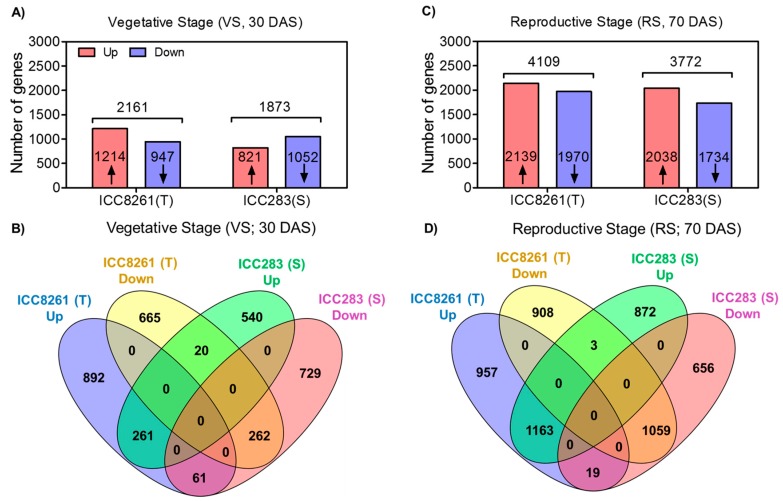
Bar graphs and Venn diagram representing genotype- and developmental stage-specific differentially expressed genes (DEGs) in response to drought stress. (**A**) DEGs up- and down-regulated in the tolerant (T, ICC8261) and sensitive (S, ICC283) genotypes at the vegetative stage (VS); (**B**) Venn diagram showing commonly up- and down-regulated DEGs amongst the genotypes at the vegetative stage (VS); (**C**) DEGs up- and down-regulated in the tolerant (T, ICC8261) and sensitive (S, ICC283) genotypes at the reproductive stage (RS); (**D**) Venn diagram showing commonly up- and down-regulated DEGs amongst the genotypes at the reproductive stage (RS).

**Figure 3 ijms-21-01781-f003:**
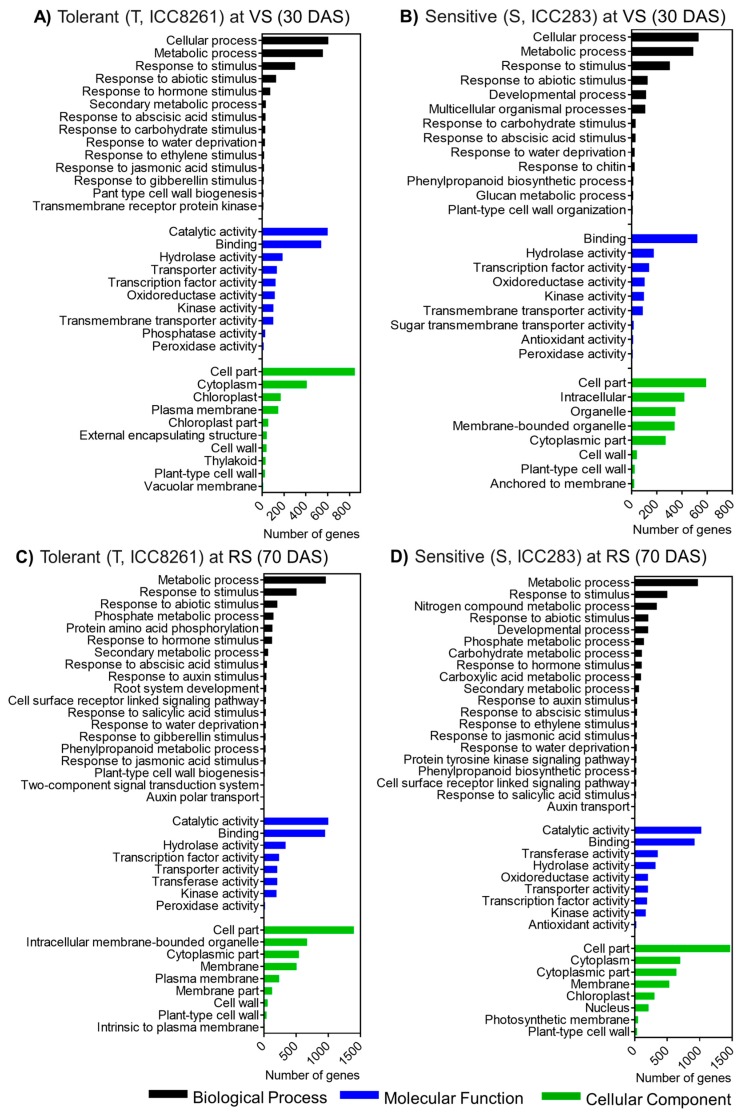
Genotype- and developmental stage-specific enrichment of gene ontology (GO) terms in (**A**) tolerant (T, ICC8261) genotype at vegetative stage (VS) (**B**) sensitive (S, ICC283) genotype at vegetative stage (VS) (**C**) tolerant (T, ICC8251) genotype at reproductive stage (RS) and (**D**) sensitive (S, ICC283) genotype at reproductive stage (RS).

**Figure 4 ijms-21-01781-f004:**
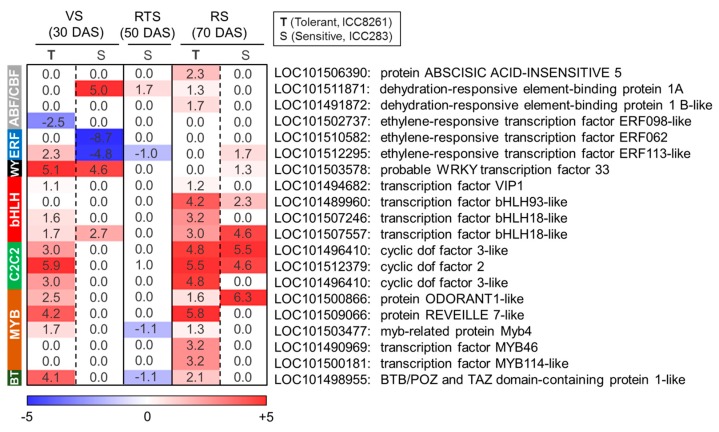
Heat map illustrating differential expression of various transcription factor (TF) gene families in the tolerant (T, ICC8261) and sensitive genotype (S, ICC283) at vegetative (VS), reproductive transition (RTS) and reproductive (RS) stages. The scale color represents log2 fold change.

**Figure 5 ijms-21-01781-f005:**
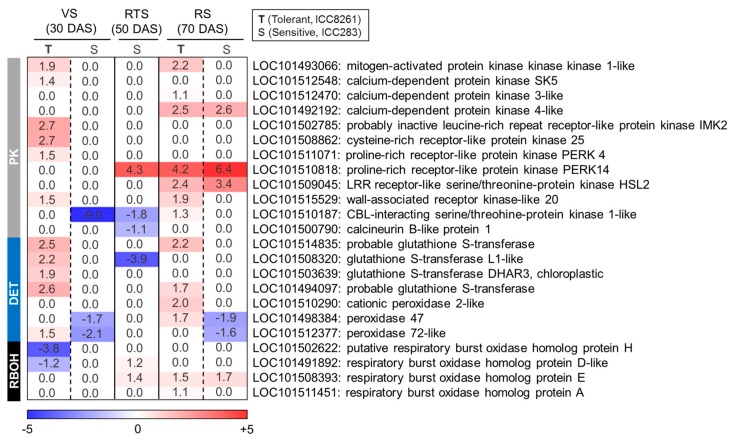
Heat map illustrating differential expression of various protein kinases (PK), detoxification enzymes (DET) and respiratory burst oxidases (RBOH) in the tolerant (T, ICC8261) and sensitive genotype (S, ICC283) at vegetative (VS), reproductive transition (RTS) and reproductive (RS) stages. The scale color represents log2 fold change.

**Figure 6 ijms-21-01781-f006:**
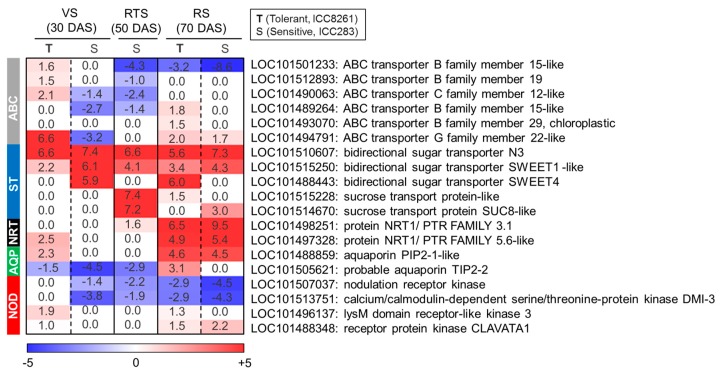
Heatmap illustrating differential expression of various transporters and root-nodule development genes that include ABC-transporters (ABC), sugar transporters (ST), NRT1/PTR transporter family (NRT1/PTR), aquaporins (AQP) and nodulation related genes (NOD) in the tolerant (T, ICC8261) and sensitive genotype (S, ICC283) at vegetative (VS), reproductive transition (RTS) and reproductive (RS) stages. The scale color represents log2 fold change.

**Figure 7 ijms-21-01781-f007:**
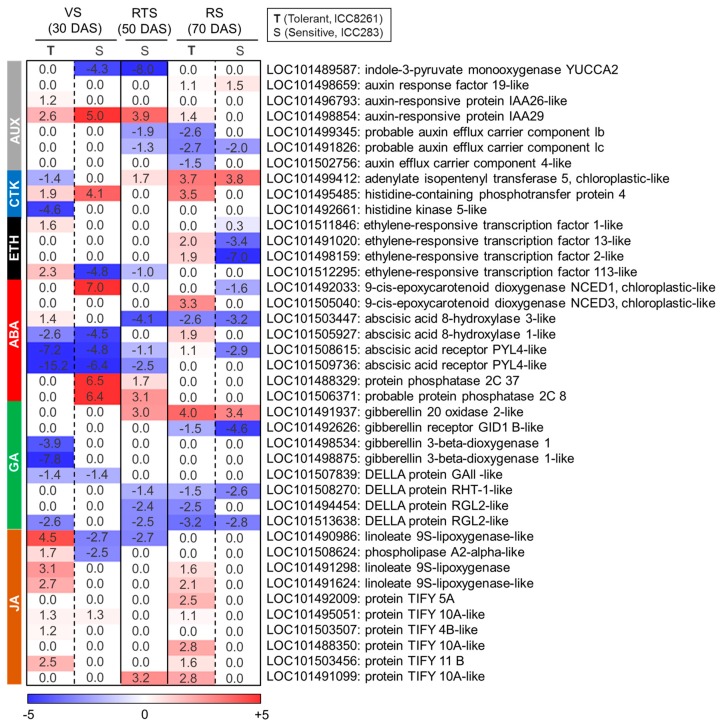
Heatmap illustrating differential regulation of auxin (AUX), cytokinin (CK), ethylene (ETH), abscisic acid (ABA), gibberellic acid (GA)and jasmonic acid (JA) related genes in the tolerant (T, ICC8261) and sensitive genotype (S, ICC283) at vegetative (VS), reproductive transition (RTS) and reproductive (RS) stages. The scale color represents log2 fold change.

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
