# Peer review of "Comparative Root Transcriptomics Provide Insights into Drought Adaptation Strategies in Chickpea (Cicer arietinum L.)"

_ijms, 2020, doi:10.3390/ijms21051781_

Round 1
Reviewer 1 Report
General comments
The article entitled “Comparative root transcriptomics provide insights on drought adaptation strategies in chickpea (Cicer arietinum L.)” focus on the transcriptomic responses of two varieties of chickpea with different drought stress tolerance, at three developmental stages. Authors have also measured several root parameters in response to drought. They found that root parameters differed across developmental stages between the tolerant and sensitive genotypes, and also that the transcriptomic responses differed between genotypes and developmental stages. The authors have associated drought stress to tolerance to increase genes involved in activities such transcriptional regulation, kinase activity, detoxification and ROS signalling.
Although I generally found that the article is interesting and well structured, I have some comments that I will point below.
1 – The Introduction is lacking more information on the physiology of the different developmental stages, and why they were chosen.
2 – The graphics with the root parameters should present the significant differences, and the P values should be provided in the text (P< 0.05, P< 0.001, etc). Also, it is stated that the SLA was lower during drought in the tolerant genotype, but the graphic (1i) is really not clear about that. Please provide P values and perhaps a table with the values and SD.
3 – Why the authors have not validated some genes by qPCR?
4- I find the Results section very long (11 pages). Please make it more concise.
Specific comments:
Introduction
Line 100-102: The sentence states the obvious! The problem during drought is that there is lack of water and problems could be avoided if water was available, obviously! Please rephrase.
Results
Line 156: “….suggest that ICC8261 shows…”
Line 164-169: Sentences are not clear. Please rephrase.
Line 171: “….regulated in these two genotypes in response to drought”.
Figure 2: Legend is incomplete and incorrect. Please check legend and correct text below.
Line 187: Gene ontology must come prior to GO, not otherwise.
Line 274: “Furthermore” instead of “Further”.
Line 283: Please add reference to the sentence.
Line 289: “Furthermore” instead of “Further”.
Line 311: “regulation” instead of “regulated”
Discussion
Line 391: Add reference to the sentence.
Line 401: please correct “geneotype”.
Line 411-412: This was only at RTS, not the other stages. Please rephrase.
Line 414-415: This is not clear in the graph.
Line 426: “…uniquely associated with each one, indicating…”.
Line 444: “was” instead of “is”.
Line 493: Add reference to the sentence.
Line 493-497. This is wrong. Only peroxidase LOC101510290 was up regulated in tolerant, others were down regulated in sensitive.
Line 511: Add reference to the sentence.
Line 513: Add reference to the sentence.
Line 549: “result” instead of “resulting”.
Line 563-564. The sentence is not clear, please rephrase.
Line 570: add “and” before “vascular system”.
Line 595: Please add reference.
Line 609: Please add reference.
Methods
4.2. Please indicate how the root parameters were measured, and how chlorophyll and SLA were determined.
Author Response
Dear Editor,
Thanks for handling our manuscript entitled “Comparative root transcriptomics provide insights on drought adaptation strategies in chickpea (Cicer arietinum L.)” Manuscript ID: ijms-711422.
We are glad that both the reviewers positively commented on our work and gave some valuable suggestions. We have revised the manuscript accordingly, and a manuscript version with track changes is provided. Also, point-by-point responses to the reviewer’s comments are included in the rebuttal letter (below, Reviewer 1).
We believe that the manuscript is much improved that can be accepted for publication in the journal.
Thank you for your consideration.
Sincerely,
Nitin Mantri,
Reviewer 1
Comment: The Introduction is lacking more information on the physiology of the different developmental stages, and why they were chosen.
Response: In the revised manuscript we have incorporated this information which read as “The effects of drought stress not only depend on duration and intensity, but also on plant developmental stage. Drought stress during the vegetative period reduces growth rate, prolongs vegetative growth and redirects root development [11-14]. Terminal drought negatively affects the formation of floral organs thus affecting fertilization and seed set formation [4, 5]. Yield losses reported in crops are greater when drought coincides with the reproductive phase [6]. Importantly, however, root-specific transcriptional responses induced by drought remain to be investigated at the developmental time-scale. The present study deals with phenotypic and transcriptomic analyses of two chickpea genotypes exhibiting tolerant (ICC8261) and sensitive (ICC283) responses to drought stress. The main objective of this work was to get a comprehensive view on drought adaptation mechanisms operating in chickpea roots by comparing genotypic- and developmental stage-specific (vegetative, reproductive transition and reproductive) responses.”
Comment: The graphics with the root parameters should present the significant differences, and the P values should be provided in the text (P< 0.05, P< 0.001, etc).
Also, it is stated that the SLA was lower during drought in the tolerant genotype, but the graphic (1i) is really not clear about that. Please provide P values and perhaps a table with the values and SD.
Response: In the revised Figures, we clearly depicted the statistically significant differences by an asterisk (*). The statistical method opted has also been described in the material and methods section under Phenotypic Analysis subsection.
Also, the statement related to SLA has been revised.
Comment: Why the authors have not validated some genes by qPCR?
Response: This data is now added as a Supplementary Figure 1. The text in the manuscript reads as “The reliability of RNA-Seq gene expression was confirmed by qRT-PCR. The fold changes obtained from RNA-seq and qRT-PCR showed good correlation with each other thus validating the RNA-Seq data (Supp. Figure 1).”
Comment: I find the Results section very long (11 pages). Please make it more concise.
Response: The revised manuscript version contains significantly shortened Result Sections by removing the redundant text.
Specific comments
Introduction
Comment: Line 100-102: The sentence states the obvious! The problem during drought is that there is lack of water and problems could be avoided if water was available, obviously! Please rephrase.
Response: This sentence has been revised now.
Results
Comment: Line 156: “…. suggest that ICC8261 shows…”
Response: Revised
Comment: Line 164-169: Sentences are not clear. Please rephrase.
Response: This sentence has been revised.
Comment: Line 171: “…. regulated in these two genotypes in response to drought”.
Response: Revised
Comment: Figure 2: Legend is incomplete and incorrect. Please check legend and correct text below.
Response: Thanks for pointing this out. We added the necessary details and corrected it.
Comment: Line 187: Gene ontology must come prior to GO, not otherwise.
Response: Revised
Comment: Line 274: “Furthermore” instead of “Further”.
Response: Revised
Comment: Line 283: Please add reference to the sentence.
Response: Revised
Comment: Line 289: “Furthermore” instead of “Further”.
Response: Revised
Comment: Line 311: “regulation” instead of “regulated”
Response: Revised
Discussion
Comment: Line 391: Add reference to the sentence.
Response: Revised
Comment: Line 401: please correct “geneotype”.
Response: Corrected
Comment: Line 411-412: This was only at RTS, not the other stages. Please rephrase.
Response: Corrected
Comment: Line 414-415: This is not clear in the graph.
Response: Revised the statement
Comment: Line 426: “…uniquely associated with each one, indicating…”.
Response: Revised
Comment: Line 444: “was” instead of “is”.
Response: Revised
Comment: Line 493: Add reference to the sentence.
Response: Revised
Comment: Line 493-497. This is wrong. Only peroxidase LOC101510290 was up regulated in tolerant, others were down regulated in sensitive.
Response: We checked it again and found that that its correct.
Comment: Line 511: Add reference to the sentence.
Response: Added
Comment: Line 513: Add reference to the sentence.
Response: Added
Comment: Line 549: “result” instead of “resulting”.
Response: Revised
Comment: Line 563-564. The sentence is not clear, please rephrase.
Response: Rephrased
Comment: Line 570: add “and” before “vascular system”.
Response: Revised
Comment: Line 595: Please add a reference.
Response: Added in the next sentence
Comment: Line 609: Please add a reference.
Response: Added
Methods
Comment: 4.2. Please indicate how the root parameters were measured, and how chlorophyll and SLA were determined.
Response: This section has been revised to detail the methods used “Phenotypic root and shoot traits were quantified in the sensitive and tolerant chickpea genotypes at three developmental stages viz. VS (30 DAS), RTS (50 DAS) and RS (70 DAS). Dry root and shoot biomass were determined after drying plant material at 65 °C for 3 days in the oven. To determine root-related traits such as length, surface area, diameter and volume harvested roots were washed with water and dried between the folds of filter paper. Then images were captured to compute these parameters. Relative water content (RWC) was calculated as RWC = [(fresh weight – dry weight)/(turgid weight – dry weight)] × 100. Leaf-related traits such as chlorophyll content (SPAD Chlorophyll Meter) and specific leaf area (SLA) was calculated as SLA = Leaf area (cm2)/Leaf dry weight (g). All the phenotypic analyses were performed in at least three replications, and data were reported as mean and standard error (SE). Means were statistically compared by one-way ANOVA at the probability level of 5% (p<0.05) by using SPSS software (IBM, New York).”
Reviewer 2 Report
Manuscript ID: ijms-711422
Type of manuscript: Article
Title: Comparative root transcriptomics provide insights on drought adaptation strategies in chickpea (Cicer arietinum L.)
Aim of this study was to compare developmental stage-specific (vegetative, reproductive transition and reproductive) phenotypic and genome-wide transcriptional responses in the roots of two chickpea genotypes showing differential sensitivity to drought stress.
A major portion of the abstract described the phenotypic data rather than transcriptomics data. Highlight the important results related to transcriptomics in abstract.
In introduction, the hypothesis/objectives of this study are not justified properly. Please justify the objectives of this study appropriately.
Discussion is too large and descriptive. Please try to make it deductive. For example, how those genes or gene families are functionally important.
It looks like the RNA sequencing data were not validated. Validate the RNA-seq data with qPCR assay.
Conclusion is too large and descriptive. A list of genes is given in conclusion. Rewrite the conclusion.
Botanical name and journal citation were not properly formatted in several cases in reference section.
Line 244: kinase kinase kinase?
Line 635: oC
Line 832: H2O2
Line 835: agronomy?
Author Response
Dear Editor,
Thanks for handling our manuscript entitled “Comparative root transcriptomics provide insights on drought adaptation strategies in chickpea (Cicer arietinum L.)” Manuscript ID: ijms-711422.
We are glad that both the reviewers positively commented on our work and gave some valuable suggestions. We have revised the manuscript accordingly, and a manuscript version with track changes is provided. Also, point-by-point responses to the reviewer’s comments are included in the rebuttal letter (below, Reviewer 2).
We believe that the manuscript is much improved that can be accepted for publication in the journal.
Thank you for your consideration.
Sincerely,
Nitin Mantri,
Reviewer 2
Comment: A major portion of the abstract described the phenotypic data rather than transcriptomics data. Highlight the important results related to transcriptomics in abstract.
Response: In the revised manuscript we put more details of the results obtained through transcriptomics, which read as “RNA-seq analysis identified genotype- and developmental-stage specific differentially expressed genes (DEGs) in response to drought stress. At the vegetative stage a total of 2161 and 1873 DEGs, and at reproductive stage 4109 and 3772 DEGs were identified in the tolerant and sensitive genotypes, respectively. Gene ontology (GO) analysis revealed enrichment of biological categories related to cellular process, metabolic process, response to stimulus, response to abiotic stress, response to hormones, etc. Interestingly, the expression of stress-responsive transcription factors, kinases, ROS signaling and scavenging, transporters, root nodulation, and oxylipin biosynthesis genes were robustly up-regulated in the tolerant genotype possibly contributing to drought adaptation. Furthermore, activation/repression of hormone signaling and biosynthesis genes was observed.”
Comment: In introduction, the hypothesis/objectives of this study are not justified properly. Please justify the objectives of this study appropriately.
Response: We revised the last paragraph of the manuscript to set proper backroad and highlight the research objectives “The effects of drought stress not only depend on duration and intensity, but also on plant developmental stage. Drought stress during the vegetative period reduces growth rate, prolongs vegetative growth and redirects root development [11-14]. Terminal drought negatively affects the formation of floral organs thus affecting fertilization and seed set formation [4, 5]. Yield losses reported in crops are greater when drought coincides with the reproductive phase [6]. Importantly, however, root-specific transcriptional responses induced by drought remain to be investigated at the developmental time-scale. The present study deals with phenotypic and transcriptomic analyses of two chickpea genotypes exhibiting tolerant (ICC8261) and sensitive (ICC283) responses to drought stress. The main objective of this work was to get a comprehensive view on drought adaptation mechanisms operating in chickpea roots by comparing genotypic- and developmental stage-specific (vegetative, reproductive transition and reproductive) responses.”
Comment: Discussion is too large and descriptive. Please try to make it deductive. For example, how those genes or gene families are functionally important.
Response: We have revised the discussion section and shortened it. Also, the role of genes is discussed in the context of their biological function to impart drought tolerance.
Comment: It looks like the RNA sequencing data were not validated. Validate the RNA-seq data with qPCR assay.
Response: This data is added as a Supplementary Figure 1. The text in the manuscript reads as “The reliability of RNA-Seq gene expression was confirmed by qRT-PCR. The fold changes obtained from RNA-seq and qRT-PCR showed good correlation with each other thus validating the RNA-Seq data (Supp. Figure 1).”
Comment: Conclusion is too large and descriptive. A list of genes is given in conclusion. Rewrite the conclusion.
Response: We have revised the conclusion. This list of key genes is given to provide a snapshot of genes possibly playing role in drought tolerance, which are not summarized elsewhere in the manuscript in one place.
Comment: Botanical name and journal citation were not properly formatted in several cases in reference section.
Response: Corrected
Comment: Line 244: kinase kinase kinase?
Response: Corrected
Comment: Line 635: oC
Response: Corrected
Comment: Line 832: H2O2
Response: Corrected
Comment: Line 835: agronomy?
Response: Revised
Round 2
Reviewer 2 Report
The last paragraph of introduction, lines 155-163, reads like results. Reduce and rewrite.
Reduce discussion further.
Author Response
Dear Editor,
Thanks for handling our manuscript entitled “Comparative root transcriptomics provide insights on drought adaptation strategies in chickpea (Cicer arietinum L.)” Manuscript ID: ijms-711422.
We have revised the manuscript as per the suggestions of the Second Reviewer, and a manuscript version with track changes is provided. Also, point-by-point responses to the reviewer’s comments are included in the rebuttal letter.
We believe that the manuscript is much improved that can be accepted for publication in the journal.
Thank you for your consideration.
Sincerely,
Nitin Mantri
Reviewer 2
Comment: The last paragraph of introduction, lines 155-163, reads like results. Reduce and rewrite.
Response: We have deleted the sentences describing results.
Comment: Reduce discussion further.
Response: We have reduced the Discussion significantly (by one and a half page).